# OpenReview forum: "Defend against Jailbreak Attacks via Debate with Partially Perceptive Agents"
_ICLR.cc/2025/Conference — ICLR 2025 Conference Withdrawn Submission_

### Official Review · Reviewer_TxXX · 2024-10-19

**Soundness:** 2
**Presentation:** 2
**Contribution:** 2
**Rating:** 3
**Confidence:** 4

**Summary:**

This paper introduces a method against jailbreak attacks on Vision Large Language Models (VLMs) that utilizes a multi-agent debate framework. The approach involves two agents: one that processes the full image and another that handles cropped images. Through debate, the agents cross-validate their outputs, allowing for effective defense—clean inputs lead to the correct output, while attacked inputs prompt re-evaluation.

**Strengths:**

The strengths of this paper include its clear and well-organized writing, making the explanation of the proposed method easy to follow. The authors provide a thorough introduction to the approach, enhancing its accessibility. Additionally, they conduct a comprehensive statistical analysis across different types of topics, which effectively demonstrates the performance of the method.

**Weaknesses:**

The weaknesses include:

- The paper claims that their method can successfully reduce the average attack success rate from 100% to 22%. However, there is no other mention of a 100% attack success rate in the main content.

- The authors compare agents with full image access and those with partial image access. At the very least, agents without image input could be a valid comparison, as the MM-SafetyBench dataset itself does not require models to analyze images. This approach is entirely feasible.

- The proposed method, which relies on debate, is resource-intensive and requires significant computational time, making it less practical. However, the paper does not discuss the efficiency of the proposed approach. Beyond defense success rates, it is unclear whether the method could lead to overly cautious responses that might affect user experience or whether it could impact the usability of the model in other multimodal tasks.

- The baselines used in the paper are overly simplistic, as they only compare the proposed method with MLLM Protector and SmoothVLM. Other relevant approaches, such as prompt-based methods, self-evaluation, self-defense, self-reminder, input perturbation techniques like query rewriting, and fine-tuning-based methods, were not included in the comparison.

- The authors' evaluation approach is relatively simple, relying solely on GPT-4 for assessment, which may introduce bias. The evaluation should include results from other methods or demonstrate consistency with human evaluation to provide a more balanced and robust assessment.

**Questions:**

Refer to Weaknesses

---

### Official Review · Reviewer_tgPe · 2024-11-03

**Soundness:** 2
**Presentation:** 3
**Contribution:** 2
**Rating:** 3
**Confidence:** 4

**Summary:**

The paper proposes a novel defense mechanism against jailbreak attacks on VLMs using a multi-agent debate approach, where one agent with full image access debates with another agent with partial/cropped image access, reducing the average attack success rate from 100% to 22% while maintaining response quality.

**Strengths:**

It seems novel to me to use multi-agent debate approach to VLM security. And the proposed method achieves significant performance improvements (reducing attack success rate from 100% to 22%) while maintaining good response quality and low refusal rates.

**Weaknesses:**

1. The paper lacks comprehensiveness in terms of attacks and defenses. Stronger attacks such as white-box attacks aren't considered. Common defenses such as refusal training aren't included.

2. Lacks computational cost comparisons between the proposed method and the baselines in the paper.

3. The claim of maintaining "quality of responses" needs more rigorous evaluation (on capability benchmarks) - the quality scoring method (0-5 scale using GPT-4) lacks detailed explanation.

**Questions:**

1. Are there any ablation studies on the impact of different components of the proposed method?

---

### Official Review · Reviewer_HjLU · 2024-11-04

**Soundness:** 3
**Presentation:** 3
**Contribution:** 2
**Rating:** 3
**Confidence:** 4

**Summary:**

This paper proposes a multi-agent debate framework focusing on defending against VLM jailbreak attacks. The framework involves 2 LLMs, one receiving the attacked image and one taking the partially observed image. The authors also propose different communication strategies to investigate the corresponding effects. Experiments show that the final conclusion reached by the two agent-debate has a low ASR against typographic attacks.

**Strengths:**

- The paper is well-written and easy to follow.
- The topic of multi-agent debate to defend VLM attacks is interesting.
- The experiments are comprehensive and clear.

**Weaknesses:**

- The novelty is limited. The multi-agent debate framework is not new and is well-explored in previous work. The authors directly apply the framework to the defense tasks.
- The paper does not fully justify the advantage of the proposed framework based on multi-agent debate. For example, to demonstrate the advantage of debating, a simple baseline can be: get multiple initial responses from the two agents in round 1 and directly do majority vote instead of debating. To demonstrate the advantage of multi-round, the author should include the comparison with: use the moderator to summarize different responses and get a final response in the first round, instead of doing multiple rounds.

**Questions:**

Please see the weaknesses above.

---

### Official Review · Reviewer_wTHj · 2024-11-12

**Soundness:** 2
**Presentation:** 2
**Contribution:** 3
**Rating:** 5
**Confidence:** 3

**Summary:**

This paper proposes a novel multi-agent debate framework for defending Vision Language Models (VLMs) against jailbreak attacks. The approach employs two types of agents - one with full image access and one with partial access - to engage in structured debates aimed at preventing harmful responses. The method achieves a significant reduction in attack success rate from 100% to 22% while maintaining better response quality than baseline approaches.

**Strengths:**

1. The training-free defense mechanism seems promising since it can be implemented at endpoints.

2. Comprehensive experimentation with multiple debate strategies like message passing, critical debate, persuasive debate.

3. The approach achieves a reduction in attack success rate from 100% to 22% while maintaining better response quality than baselines.

**Weaknesses:**

1. Experimental Design Issues:


- Insufficient sample size: only 20 samples per scenario for experiment is not adequate enough

- Inadequate justification for baseline selection: there is no explanation in the article why those two baseline methods are chose for comparison.


2\. Evaluation methodology Concerns:

- Unclear indicator function definition: the article didn’t clearly explain what the indicator function actually means

- No explicit criteria for "successful" attacks: there is no explicit definition of what constitutes a successful attack

- Over-reliance on GPT-4 without human validation: there is no human evaluation to validate GPT4’s assessments, which introduces no clear criteria for quality scoring


3\. Grammar errors, for example

1. ... other debater's answers -> ... other debaters' answers

2.  ... can notebly decrease... -> ... can notably decrease...

3. ... Additionlly ... -> Additionally...

**Questions:**

see weakness.

---

### Note · Authors · 2024-11-14

I have read and agree with the venue's withdrawal policy on behalf of myself and my co-authors.